# Characterization of BoHV-1 gG-/tk-/gE- Mutant in Differential Protein Expression, Virulence, and Immunity

**DOI:** 10.3390/vetsci8110253

**Published:** 2021-10-27

**Authors:** Marawan A. Marawan, Mingliang Deng, Chen Wang, Yingyu Chen, Changmin Hu, Jianguo Chen, Xi Chen, Huanchun Chen, Aizhen Guo

**Affiliations:** 1State Key Laboratory of Agricultural Microbiology, Huazhong Agricultural University, Wuhan 430070, China; dml050806001@gmail.com (M.D.); wangchen2012@webmail.hzau.edu.cn (C.W.); chenhch@mail.hzau.edu.cn (H.C.); 2College of Veterinary Medicine, Huazhong Agricultural University, Wuhan 430070, China; chenyingyu@mail.hzau.edu.cn (Y.C.); hcm@mail.hzau.edu.cn (C.H.); chenjg@mail.hzau.edu.cn (J.C.); chenxi@mail.hzau.edu.cn (X.C.); 3Infectious Diseases, Animal Medicine Department, Faculty of Veterinary Medicine, Benha University, Toukh 13736, Egypt; 4Hubei International Scientific and Technological Cooperation Base of Veterinary Epidemiology, Key Laboratory of Preventive Veterinary Medicine in Hubei Province, Wuhan 430070, China; 5Key Laboratory of Development of Veterinary Diagnostic Products, Ministry of Agriculture, Huazhong Agricultural University, Wuhan 430070, China; 6Key Laboratory of Ruminant Bio-Products of Ministry of Agriculture and Rural Affairs, Huazhong Agriculture University, Wuhan 430070, China

**Keywords:** bovine herpesvirus 1,5, proteomics, glycoprotein, marker vaccines, virulence, immunity

## Abstract

Infectious bovine rhinotracheitis (IBR), caused by bovine alphaherpesvirus 1 (BoHV-1), is an important disease affecting cattle worldwide resulting in great economic losses. Marker vaccines are effective in controlling infectious diseases including IBR, because they allow the discrimination between the natural infection and the vaccination. Therefore, a triple gene deleted strain BoHV-1 gG-/tk-/gE- was developed and evaluated in vivo and in vitro as a marker vaccine. In cell culture, this triple mutant virus showed significantly slower growth kinetics and smaller plaques when compared to wild-type (wt) BoHV-1 and double mutant BoHV-1 gG-/tk- (*p* < 0.01). On proteomic level, it revealed downregulation of some virulence related proteins including thymidine kinase, glycoproteins G, E, I, and K when compared to the wt. In vitro, the triple mutant virus showed a significantly lower and shorter viral shedding period (*p* < 0.001) in calves compared to double mutant. Moreover, the immunized calves with triple mutant virus showed protection rates of 64.2% and 68.6% against wt BoHV-1 and wt BoHV-5 challenge, respectively, without reactivation of latency after dexamethasone injection. In conclusion, BoHV-1 gG-/tk-/gE- is a safer marker vaccine against IBR although its immunogenicity in calves was decreased when compared to double mutant virus.

## 1. Introduction

Bovine alphaherpesvirus-1 (BoHV-1), the causative agent of infectious bovine rhinotracheitis disease (IBR) is a *Varicellovirus* genus member that belongs to subfamily *Alphaherpesvirinae* under the family *Herpesvirida* [1]. *Alphaherpesviruses* are double-stranded DNA viruses and their genome is at least 120 kb long, encoding for 70 or more genes [2].

Three antigenically similar subtypes of BoHV-1 have been revealed depending on the genomic analysis and viral peptide patterns namely, BoHV-1.1, BoHV-1.2a, and BoHV-1.2b [3]. Furthermore, the disease outcome based on the BoHV-1 subtypes in which BoHV-1.1 is the most predominant and mainly culminated in IBR with respiratory signs whilst BoHV-1.2 which can be divided into BoHV-1.2a and BoHV-1.2b has a broad range of clinical manifestations including IBR, infectious pustular vulvovaginitis, and balanoposthitis [4,5]. On the other hand, BoHV-5 is neuropathogenic and responsible for fatal meningoencephalitis [6]. The long-life latency development in the sensory ganglia and the existence of carrier animals are pivotal and serious features of BoHV-1 infections that render the control of the disease not feasible due to induction of reactivation from latency that was mimicked by various stressors particularly the synthetic corticosteroid dexamethasone [2,7].

Vaccination is the most successful way to control and eradicate IBR with some disadvantages related to the types of vaccines, either the inactivated and or attenuated IBR conventional vaccines. To overcome these disadvantages, genes that are virulence-related but non-essential for viral viability are the targets to be deleted for novel vaccine preparation [8,9]. Commonly used genes in development of gene-deleted vaccines to IBRV are from glycoprotein mutants such as Envelope glycoprotein E (gE), Envelope glycoprotein G (gG), and Envelope glycoprotein N (gN), and some genes coding for important enzymes such as Thymidine kinase (tk) [10,11]. Further, multiple deletion in one construct is considered to decrease the virulence and risk of the back-mutation such as gE-/tk- [12,13], gE-/gG-/Tegument protein US2- [14], and gN-30-32CT-null/gE-CT-/Envelope protein US9- [15]. On the other hand, proteins are the core mediators of viral functions and any abnormal alteration on their abundance or expression levels may reflect the modification in viral pathological processes and immunity inside the hosts [16]. Usually attenuation of viral virulence is associated with decrease of viral immunogenicity.

Previously, our group has successfully constructed and characterized a double deleted strain BoHV-1 gG-/tk-, which was attenuated in calves and yet maintained the ability to stimulate a protective immune response [17]. Based on it, the construction and characterization of triple mutant BoHV-1 gG -/tk-/gE- was carried out and differential protein expression investigated by proteomic analysis using label-free quantitative proteomics (LFQP). Then, in vivo evaluation of the virulence and protective efficacy of this triple mutant strain against homologous BoHV-1 and heterologous BoHV-5 challenge in calves was performed to determine its potential application as a novel IBR marker vaccine.

## 2. Materials and Methods

### 2.1. Ethics Statement

The protocols regarding animal experiments were approved by the Committee on the Ethics of Animal Experiments at Huazhong Agricultural University and conducted in strict accordance with the Guide for the Care and Use of Laboratory Animals, Wuhan, Hubei, China.

### 2.2. Cell Culture and Virus Strains

Madin-Darby Bovine Kidney (MDBK) cell line was used for virus propagation. The cells were cultured with the complete medium Dulbecco’s Modified Eagle’s Medium (DMEM) containing 10% fetal calf serum (FCS), 100 µg/mL streptomycin, and 100 IU/mL penicillin in a humidified incubator at 37 °C and 5% CO_2_ as described previously [18].

Three BoHV-1 strains were used: (i) wt BoHV-1 of bovine origin isolated from a diseased calf with a respiratory sign by our laboratory. (ii) The wt BoHV-5 strain was kindly provided by Fabrício Campos at Federal University of Rio Grande doSul (UFRGS), Porto Alegre, Brazil. Finally, (iii) double mutant strain BoHV-1 gG-/tk-, a vaccine strain with deleted gG and tk, was constructed and characterized by this lab as reported previously [19]. The three strains at a titer of 10^7^ PFU/mL (plaque forming unit/mL) were stored as viral stocks and used for MDBK cell infection with DMEM containing 2% FCS.

### 2.3. Construction of Triple Mutant Virus BoHV-1 Strain gG-/tk-/gE-

#### 2.3.1. Construction of Transfer Vector pBoHV-1 gE-

The primer pairs P1/P2 with Hind III/Kpn I sites and P3/P4 with BamH I/EcoR I sites were designed to amplify gE-upstream homologous arm and downstream homologous arm, respectively (Appendix A). Similarly, the primer pair P5/P6 with KpnI/BamHI sites were used to amplify the enhanced green fluorescent protein (egfp) gene expression cassette. The resultant gE upper arm, egfp gene, and gE lower arm were cloned sequentially into pcDNA3.1 (+) myc-HisB to obtain recombinant plasmid pBoHV-1 gE- (Appendix A).

#### 2.3.2. Generation and Characterization of the Triple Mutant Virus

The linear plasmid pBoHV-1 gE- and full-length BoHV-1 gG-/tk- viral genomic DNA (the plasmid was cut firstly by restriction enzyme Hind III and BamH I) was purified and then co-transfected into MDBK cells for homologous recombination by using the calcium phosphate method as described previously [17]. The putative recombinant viral plaques were picked and subsequently purified for five rounds. The mutants were analyzed subsequently to determine the deletion of gE gene by various strategies of PCR using the primers (Appendix A) specific to different fragments of the gE gene region and the PCR products were sequenced for confirmation (Sangon Biotech, Shanghai, China) (Appendix A).

Four pairs of primers including P7/P8 (Lanes 1–4), P11/P12 (Lanes 5–8), P3/P14 (Lanes 9–12), and P7/P10 (Lanes 13–16) were used for identification of wt BoHV-1 and difference gene-deleted mutants (Figure 1A). Moreover, three pairs of primers including P7/P8 (Lanes 1–3), P9/P10 (Lanes 4–6), P7/P10 (Lanes 7–9) were used for further identification of BoHV-1 gG-/tk-/gE- mutant. Each pair of primers set has three repeats (Figure 1B).

### 2.4. Viral Growth Kinetics and Plaque Size Determination

For growth characteristics comparison, the plaque morphologies and one-step growth curves of the three viruses were determined and compared as described elsewhere [19].

### 2.5. Label Free Quantitative Proteomics (LFQP)

#### 2.5.1. Virions Purification

Viral purification was carried out using the sucrose density gradient ultracentrifugation method (Appendix A). The resultant viral pellets were resuspended in 200 μL 1× TE buffer (10× TE buffer (1000 mL): 20 mL 500 mM/L EDTA (pH 8.0) and 100 mL 1 M/L Tris HCl buffer (pH 8.0) were added to water and final volume was 1000 mL. 1× TE buffer was prepared by diluting 10× TE buffer. Finally, 5 mL pure virus in TNE of each strain were obtained. Meanwhile the OD260/OD280 ratio was measured to be 0.5, 0.4, and 0.5 for wt, double mutant, and triple mutant, respectively, and the total protein concentration was calculated to be 3526.6, 2728.0, and 3370.8 μg/mL. The viral products were divided into several aliquots and kept at −80 °C for further use.

#### 2.5.2. LC-MS/MS Analysis

LC-MS/MS analysis was performed on 500 μL viral products (Project# Z9358LQ, PTM-Biolabs Co., Ltd., Hangzhou, Zhejiang, China). Briefly, the viral samples experienced protein extraction and quantification, and trypsin digestion [20]. Then the resultant tryptic peptides were dissolved and fractionated using UPLC system and subjected to NSI source followed by tandem mass spectrometry (MS/MS) in Q Exactive ^TM Plus^ (Thermo Scientific ^TM^ Orbitrap Fusion Lumos ^TM^, Hangzhou, Zhejiang, China). The fixed first mass was set as 100 *m*/*z* (Appendix A). The MS data validation is shown in Appendix A and was carried out using peptide mass tolerance distribution (Appendix A), identified peptide length distribution (Appendix A), and protein mass and coverage distribution (Appendix A).

#### 2.5.3. Database Search and Bioinformatic Analysis

The resulting MS/MS data were processed using the Maxquant search engine (v.1.5.2.8 http://www.maxquant.org, access on 20 September 2020). The spectral data were searched against the target BoHV-1 Bos Taurus referenced protein database downloaded from UniProt (Universal Protein Resource) (www.uniprot.org, access 25 October 2021). Bioinformatic analysis was performed using Gene Ontology (GO) annotation proteome and was classified based on three categories: biological process, cellular component, and molecular function. In addition, a 1.5-fold-change was used as the threshold of differential expression change, and the statistical *t*-test with *p* < 0.05 was used as the threshold of significance. Further, the quantitative ratio over 1.5 was considered upregulation while the quantitative ratio less than 1.5 as downregulation.

### 2.6. Cattle Experiments

Virulence and protective efficacy of the triple deletion mutant were evaluated in two-month-old weaned male Holstein calves that were seronegative for BoHV-1 (*n* = 24) (purchased from Hubei Center of Disease Control). The animal experimental design is summarized in Table 1. Each group of calves was housed in a separate room to prevent intergroup transmission. The animals were anesthetized by spray with 10% Lidocaine on the nasal cavity of the calves before the infection, vaccination, and challenge which was as used by Valera et al. [21]. Nasal swabs were collected daily in 2 mL of tissue culture medium supplemented with 2% penicillin and streptomycin for 14 days following virus exposure and dexamethasone injection. Blood was collected weekly up to the end of the experiments. The samples were processed and stored at −80 ℃. The animals were euthanized at the end of the experiments and the lung tissues were sampled for pathological examination.

#### 2.6.1. Clinical Evaluation

The animals’ body temperature, behavior, presence of coughs, abnormal respiration, ocular and nasal discharges, hyperemia or lesions of the nasal mucosa, and conjunctivitis were monitored and recorded daily, and clinical scores were assigned for each parameter as described earlier [22]. Neurological signs were recorded in the wt BoHV-5 challenge experiment. The daily clinical score of each calf was the sum of the scores of all clinical parameters. The mean daily clinical score was calculated for each group and compared among groups.

#### 2.6.2. Virus Isolation and Titration

Nasal swabs were used for virus isolation and titrations as described previously [23].

#### 2.6.3. Serological Investigation

Serum samples were collected at different time points after vaccination or challenge and submitted to a standard virus-neutralizing (VN) assay as described earlier [23]. Sera were also submitted to BoHV-1 gB-specific antibody test (IDEXX Laboratories, USA), BoHV-1 gE-specific antibody test (IDEXX Laboratories, Westbrook, ME, USA), sIgA (Bethyl Laboratories, Montgomery, TX, USA), and serum cytokine tests including IFN-γ (Mabtech, Nacka Strand, Sweden), IL-2 (Ray Biotech, Norcross, GA, USA), and IL-4 (Mabtech, Nacka Strand, Sweden) detection by using commercial ELISA kits.

#### 2.6.4. Histo-Pathological Examination of Lungs

The calves were euthanized 28 dpc and a 35-point scoring system was used for the lesion evaluation in each lung lobe as described previously [24].

The lesion evaluation in each lung lobe was as follows: 0, indicated no visible lesions; 1, no gross lesions, but lesions were apparent upon dissection; 2, <5 gross lesions with diameters of <10 mm; 3, >6 gross lesions with diameters of <10 mm, or a single distinct gross lesion with a diameter of >10 mm; 4, 2 or more distinct gross lesions with diameters of >10 mm; 5, gross coalescing lesions. The scores of the individual lobes were summed to generate the total lung score.

### 2.7. Statistical Analysis

The protection rate was calculated according to the formula described by Zhang et al. [25]. The data for the various groups were compared using *t*-tests and a one-way ANOVA. Results of comparisons with *p* < 0.05 (*) or *p* < 0.01 (**) or *p* < 0.001 (***) were considered to indicate a significant or high significant statistical difference.

## 3. Results

### 3.1. Construction and Characterization of the Recombinant BoHV-1 gG-/tk-/gE- Triple- Deleted Mutant

The BoHV-1 gG-/tk-/gE- virus was confirmed by PCR (Figure 1). A fragment of 2395 bp (primers P7/8) including the whole egfp gene and parts of the flanking upstream and downstream sequences (Figure 1A, Lane 3), and a fragment of 481 bp (primers P7/10) including parts of upstream homologous gE arm and egfp gene (Figure 1A, Lane 15) were amplified from the triple mutant, respectively. The deleted gG and tk gene fragments in BoHV-1 gG-/tk-/gE- were also detected (Figure 1A, Lanes 7 and 11, respectively). Additionally, a fragment of 1685 bp, including the whole upstream homologous gE arm gene, parts of the flanking upstream sequences of the upstream homologous gE arm, and egfp gene were amplified from the triple mutant (Figure 1B, Lanes 4–6). The PCR products of the mutant were custom-sequenced and the viral gE gene was found to be replaced with egfp gene.

### 3.2. Comparative Viral Growth Kinetics and Plaque Size Estimation

Compared to wt BoHV-1 and BoHV-1 gG-/tk-, BoHV-1 gG-/tk-/gE- showed a weaker cell cytopathic effect (CPE) in MDBK cells shown by smaller plaques with uncleared edges within 48 h post-infection (hpi) (Figure 2A). The mean diameter of BoHV-1 gG-/tk-/gE- plaques was 308 ± 20 μm, 48 hpi, which was significantly less than wt BoHV-1 and the BoHV-1 gG-/tk- mutant (*p* < 0.001) (Figure 2B). The growth curves of these three viruses were similar during the first 6 h, however, the amount of BoHV-1 gG-/tk-/gE- virus produced was less than the wt BoHV-1 virus and BoHV-1 gG-/tk- mutant at the remaining time points (*p* < 0.001) until 48 hpi (Figure 2C).

### 3.3. Comparative Proteomics of the Three BoHV-1 Strains

#### 3.3.1. Differentially Expressed Proteins in Different Comparable Strains

A total of 69 protein groups were identified, among which 65 proteins were accurately quantified covering almost all the viral proteins with a quantification ratio consistent in at least two of the three LC-MS/MS analyses (Appendix A). Further, the differentially expressed proteins were assessed.

Compared to wt BoHV-1, BoHV-1 gG-tk-gE- triple mutant exhibited downregulation of 16 proteins (namely, gG, TK, gE, gI, gK, DNA replication helicase, BICP4, DNA primase, US1.67, UL7, UL14, UL20, UL41, UL43, UL46, UL51) and upregulation of 12 proteins (gB, gD, UL6, UL24, Triplex capsid protein 1, Triplex capsid protein 2, US3 virion serine/threonine protein kinase, Major capsid protein, Nuclear egress protein 1, Capsid scaffolding protein, Alkaline nuclease, Small capsomere-interacting protein). On the other side, when compared with the double mutant, BoHV-1 gG-tk-gE- triple mutant exhibited 11 downregulated proteins (gE, DNA replication helicase, DNA primase, gI, gK, UL20, UL46, circ protein, virion host shutoff factor, nuclear egress protein 2 and cytoplasmic envelopment protein 1) and 24 proteins were upregulated (gB, gC, gD, gL, gM, gN, α-TIF, UL3, UL3.5, UL4, UL15, UL43, Vp8, UL50, US1.67, Triplex capsid protein 1, Triplex capsid protein 2, Portal protein, Uracil-DNA glycosylase, BICP0, Major capsid protein, Capsid scaffolding protein, Tripartite terminase subunit 3, Alkaline nuclease, Small capsomere-interacting protein).

Furthermore, the BoHV-1 gG-tk- douple mutant revealed downregulation of 21 proteins (gG, TK, gD, gL, gC, gN, gM, BICP4, Alpha TIF protein, UL3, UL3.5, UL4, UL14, UL15, UL41, UL43, UL46, VP8, UL50, UL51 and US 1.67 protein) and upregulation of one protein only (Major DNA binding protein) when compared to the wt BoHV-1.

Surprisingly, in correlation to the wt BoHV-1, both the double and triple mutants have (a) nine common downregulated proteins (gG, TK, BICP4, UL14, UL41, UL43, UL46, UL51, and US1.67); (b) 12 uniquely upregulated proteins in the double mutant (gC, gD, gL, gM, gN, α-TIF, UL3, UL3.5, UL4, UL15, Vp8, and UL50); (c) seven uniquely downregulated in the triple mutant (gE, DNA replication helicase, DNA primase, gI, gK, UL7, and UL20) (Appendix A).

#### 3.3.2. GO Secondary Annotation Classification

GO classification using the GO Terms Level 2 database was used to examine the traits of the differentially expressed proteins regarding their biological processes (BPs), cellular components (CCs), and molecular functions (MFs) (Figure 3). Comparing the wt BoHV-1 to double mutant BoHV-1 gG-/tk- revealed that most of the proteins were involved in the cellular process (32%); followed by equal involvement in metabolic process, multi organism process and biological regulation (18%); then, similar involvement in localization and cellular components organization or biogenesis (5%) and single organism process (4%). By cellular components, the classification showed that most of the proteins were located in the membrane (55%), other organisms (36%) and virions (9%). Meanwhile, classification by molecular functions demonstrated that most of the proteins were involved in binding (73%) and catalytic activity (27%).

Comparison of wt BoHV-1 to the triple mutant BoHV-1 gG-/tk-/gE- in biological process demonstrated that the differentially expressed proteins were involved in the cellular process (34%), metabolic process (23%), multi organism process (19%); then, equal involvement in biological regulation and cellular components organization or biogenesis (8%); similar involvement in locomotion and single organism process (4%). By cellular components, the classification showed that most of the proteins belonged to the membrane (41%), other organisms (30%), and virions (29%). By molecular functions, the proteins were involved in binding activity (47%), catalytic activity (41%), and structure molecule activity (12%).

Finally, comparison of BoHV-1 gG-/tk- to the triple mutant BoHV-1 gG-/tk-/gE- in biological process demonstrated that the differentially expressed proteins were involved in the cellular process (40%), multi organism process (30%), metabolic process (20%), and biological regulation (10%). By cellular components, the classification showed that most of the proteins were located in the membrane (71%) and other organisms (29%). While binding and catalytic activities of differentially expressed proteins shared the half percentage of molecular functions (50%).

### 3.4. In Vivo Experiment Findings

#### 3.4.1. Virulence and Reactivation of BoHV-1 gG-/tk-/gE- in Calves

Following inoculation of calves, BoHV-1 gG-/tk-/gE- did not induce fever (Appendix A) or clinical signs (Appendix A). In contrast, the calves inoculated with wt BoHV-1 displayed clinical signs, including coughing, ocular and nasal discharges, conjunctivitis, depression, and abnormal breathing (Appendix A), and their clinical scores were significantly higher than that of the other groups (*p* < 0.001) (Appendix A).

In addition, both mutant-inoculated groups showed significantly less viral shedding (*p* < 0.001) in nasal swabs and shed for a shorter period than the group inoculated with wt BoHV-1 (Appendix A and Appendix A). The BoHV-1 gG-/tk- inoculated calves shed virus for 6.3 ± 1.4 days with a peak viral titer of 10^2.83^ PFU/mL at 4 dpi, whereas the BoHV-1 gG-/tk-/gE- inoculated calves shed virus only for 3.5 ± 0.8 days, with a peak viral titer of 10^2.04^ PFU/mL at 4 dpi, and showed significantly less viral shedding (*p* < 0.001) than the former group (Appendix A).

Neither the BoHV-1 gG-/tk-/gE- nor the BoHV-1 gG-/tk- mutant could be reactivated by dexamethasone injection in calves. However, the wt BoHV-1 was detected in nasal swabs 1–3 days after the dexamethasone injection (Appendix A)

#### 3.4.2. Protection of BoHV-1 gG-/tk-/gE- against wt BoHV-1 Challenge in Calves

To evaluate the protective efficacy of BoHV-1 gG-/tk-/gE- against wt BoHV-1 challenge, three calves from each group were challenged with wt BoHV-1. Two calves in BoHV-1 gG-/tk-/gE- inoculated group and one calf in BoHV-1 gG-/tk- inoculated group experienced fever (>39.7 ℃). All calves in the unvaccinated control group had elevated temperatures above 40 ℃ for 3 days (Figure 4A). The number of days of fever, ocular and nasal lesions, and cough are presented in Appendix A. The clinical scores recorded after the challenge are shown in Figure 4B. One of three calves in both mutant vaccinated groups displayed clinical signs with a score of more than 3 after the challenge. All calves in the unvaccinated control group displayed fever, nasal and ocular discharge, cough, and conjunctivitis. As a result, their clinical scores were significantly increased after the challenge (Figure 4B). The negative control group did not display any abnormal clinical signs.

The calves in BoHV-1 gG-/tk-/gE- inoculated group shed virus for 9.3 ± 1.1 days, with a peak viral titer of 10^4.18^ PFU/mL at day 5 dpc. Compared to the BoHV-1 gG-/tk-/gE- inoculated group, calves in the BoHV-1 gG-/tk- inoculated group shed the virus for a shorter period (6.0 ± 1.0 days) with a peak viral titer of 10^3.18^ PFU/mL at 5 dpc with less, although not significant, viral shedding (*p* > 0.05). The unvaccinated calves shed virus for 11.3 ± 1.5 days with a peak viral titer of 10^5.32^ PFU/mL at 5 dpc (Appendix A), which was significantly higher (*p* < 0.01) than those of both groups of mutant vaccination (Figure 4C).

The mean score for lung pathology per calf for the BoHV-1 gG-/tk-/gE-, BoHV-1 gG-/tk-, unvaccinated but challenged, and blank control groups was 6.00 ± 3.00, 2.67 ± 2.08, 13.33 ± 3.06, and 2.00 ± 1.00, respectively (Figure 4D). The mean score of the unvaccinated group was significantly higher than those of the other groups (*p* < 0.01).

The protection rates were calculated based on the totals of the various scoring systems (Appendix A). The overall protection rate for the BoHV-1 gG-/tk-/gE- and BoHV-1 gG-/tk- group was 91.7% and 64.2%, respectively.

#### 3.4.3. Cross-Protection of BoHV-1 gG-/tk-/gE- against wt BoHV-5 Challenge in Calves

To check whether vaccination with BoHV-1 gG-/tk-/gE- or BoHV-1 gG-/tk- could induce cross-protection to BoHV-5 infections, three calves from each group were challenged with wt BoHV-5 strain. After challenge, one calf in both groups (BoHV-1 gG-/tk-/gE- and BoHV-1 gG-/tk-) experienced fever (>39.7 ℃) for 1 day. In contrast, two calves in the unvaccinated group had a temperature above 40 ℃ for 3 and 4 days, respectively (Figure 5A). Respiratory signs detected in both mutant-inoculated groups were less severe and remained for a shorter period than those observed in unvaccinated animals (Appendix A). The clinical scores recorded after the challenge are shown in Figure 5B. Neurological signs including head-pressing were observed in one calf of the BoHV-1 gG-/tk-/gE- inoculated group for 3 consecutive days (8–10 dpc) and in one calf of the BoHV-1 gG-/tk- inoculated group for 1 day (7 dpc), while head-pressing was observed in two calves of the unvaccinated groups for 3 consecutive days.

Unvaccinated calves shed virus for 12.0 ± 1.7 days with a peak shedding at 6 dpc (10^4.06^ PFU/mL). However, the viral shedding in BoHV-1 gG-/tk-/gE- and BoHV-1 gG-/tk- inoculated groups was observed for 9.3 ± 1.5 and 8.0 ± 1.0 days, respectively. The peak viral titer of BoHV-1 gG-/tk-/gE- and BoHV-1 gG-/tk- was 10^3.25^ (6 dpc) and 10^2.88^ PFU/mL (4 dpc), respectively (Appendix A). In addition, viral titers shed from the BoHV-1 gG-/tk-/gE- and BoHV-1 gG-/tk- inoculated groups were significantly lower than that of the unvaccinated group at 6 dpc (BoHV-1 gG-/tk-/gE- vs. unvaccinated: *p* < 0.05, BoHV-1 gG-/tk- vs. unvaccinated: *p* < 0.01) (Figure 5C).

The mean score for lung pathology for the BoHV-1 gG-/tk-/gE-, BoHV-1 gG-/tk-, unvaccinated but challenged, and blank control groups was 6.00 ± 4.58, 9.00 ± 6.00, 15.33 ± 7.57, and 2.00 ± 1.00, respectively (Figure 5D). The mean score for the unvaccinated group was higher than that of the blank control group (*p* < 0.05) but without significant differences while compared with that of the other groups.

The protection rates were calculated based on the total scores of the various scoring systems (Appendix A). The overall protection rate for the BoHV-1 gG-/tk-/gE- and BoHV-1 gG-/tk- group was 68.6% and 47.22%, respectively.

### 3.5. Serological Investigations

#### 3.5.1. Humoral Immune Response in Calves

The serum samples obtained at intervals after vaccination and challenge were tested for VN antibody (Appendix A). Two of the six calves in the BoHV-1 gG-/tk-/gE- inoculated group were positive for VN antibody at 28 dpi, and four calves were positive at 35 dpi. In contrast, two of six calves in the BoHV-1 gG-/tk- inoculated group were positive at 14 dpi and five calves were positive at 21 dpi. In addition, there were significantly lower (*p* < 0.01) VN antibody titers in the BoHV-1 gG-/tk-/gE- inoculated calves (0.8 ± 1.3) than in the BoHV-1 gG-/tk- inoculated group (6.0 ± 4.5) at 28 dpi (Appendix A). Interestingly, following the wt BoHV-1 challenge, a strong and rapid increase of VN antibody titers was observed in both inoculated groups. Similarly, the titers in the BoHV-1 gG-/tk-/gE- inoculated group increased, although the titers were significantly lower (*p* < 0.01) at 56 dpi when compared to the BoHV-1 gG-/tk- inoculated group (Appendix A). Moreover, following wt BoHV-5 challenge, the calves in both the mutant-inoculated groups developed VN antibodies to BoHV-1 (Appendix A) and BoHV-5 (Appendix A), and the BoHV-1 gG-/tk-/gE- inoculated group also had similar increases when compared to the BoHV-1 gG-/tk- inoculated group. Further, both mutant-inoculated groups developed VN antibodies with no significant differences in VN antibody titers to wt BoHV-1 virus (Appendix A) and wt BoHV-5 virus (Appendix A) after wt BoHV-5 challenge. In contrast, the unvaccinated control calves developed a lower and delayed antibody response after challenge, which were typical of primary responses.

Sera from all calves of the BoHV-1 gG-/tk- and wt BoHV-1 inoculated groups had positive (≥55%) anti-BoHV-1 gB antibodies at 14 dpi, while 21 dpi for BoHV-1 gG-/tk-/gE- inoculated group (Appendix A). Groups that were BoHV-1 gG-/tk- vaccinated or wt BoHV-1 infected were found to be positive for gE-specific antibodies at 21 dpi. However, no anti-gE antibody was detected in all calves of the BoHV-1 gG-/tk-/gE- inoculated group before the challenge (Appendix A).

IgA increased in the sera of BoHV-1 gG-/tk-/gE- inoculated calves after vaccination, although this was not significantly different from that of the BoHV-1 gG-/tk- and wt BoHV-1 inoculated groups (Appendix A).

#### 3.5.2. Cytokine Production in Calves

IFN-γ was detected but none of the BoHV-1 mutant vaccinated groups generated a significant increase in the level (Appendix A). IL-2 was not detectable from any calf. IL-4 was detected in all calves of the BoHV-1 gG-/tk- inoculated group at 1 dpi, whereas in the BoHV-1 gG-/tk-/gE- inoculated group only two calves were detectable after vaccination (Appendix A), and the level was significantly lower (*p* < 0.05) than that of the BoHV-1 gG-/tk- inoculated group at 14 dpi. However, none of the wt BoHV-1 infection groups were detectable in IL-4 after infection.

## 4. Discussion

Vaccination is an effective control measure against IBR. Several European countries have initiated control programs aimed at BoHV-1 eradication based on the use of marker vaccines [26]. These marker vaccines have one or more antigenic proteins less than the parental wt virus, and it is possible to detect an antibody response to the specific deleted protein which could allow differentiation of infected from vaccinated animals [27].

To diminish the pathogenicity and avoid virus recombination, triple mutant BoHV-1 gG-/tk-/gE- was developed on the base of the double mutant vaccinal strain BoHV-1 gG-/tk as a superior vaccine candidate [28]. On the other side, LFQP helps control viral infectious disease affecting cattle via identification of virion-associated proteins and it can translate the basic science investigations into practical measures [29,30].

The BoHV-1 gG-/tk-/gE- mutant exhibited growth kinetics inferior to that of wt BoHV-1, and double mutant gG-/tk- with smaller plaques of uncleared edges and reduced sizes observed during the in vitro study. Moreover, it showed a significantly lower virus shedding and shorter period of shedding in vivo. Therefore, BoHV-1 gG-/tk-/gE- exhibited more attenuation and less virulence in vivo and in vitro which might be attributed to missing the key role of gE in cell-to-cell spread of the virus in cultured cells and virulence [31,32,33,34].

From a proteomic point of view, the GO classification was mainly involved in the cellular process and the metabolic process by BPs belongs to membrane part by CPs and exhibited binding and catalytic activities by MFs. This attributed to the virion assembly of gG and gE as a membrane envelope protein and their role in cell-to-cell spread, cell attachment, and chemokine binding function as well as the crucial effect of tk as a virulence-related gene [18,19].

The BoHV-1 gG-/tk-/gE- mutant appeared to be immunogenic and less virulent comparing to other virions due to the deletion of virulence-related genes (tk and gE) and immunosuppressive gG [19,35]. These results indicate the triple gene deleted virions lost the greatest part of its virulence comparing to the other two virions with retaining its immunogenicity [17,36,37,38].

After BoHV-1 challenge, the BoHV-1 gG-/tk-/gE- could still protect against virulent wt BoHV-1 challenge even though it was more attenuated. However, the protection against virulent BoHV-1 challenge of BoHV-1 gG-/tk-/gE- was lower than that of BoHV-1 gG-/tk- since the calves inoculated with BoHV-1 gG-/tk-/gE- show higher clinical scores, a shorter period of virus shedding. The reason for lower protection might be that the immunogenicity of BoHV-1 strains is directly related to their replication efficiency in vivo [39]. Hence, gene deletions for attenuation may affect replication and, consequently, compromise the immunogenicity [40]. Thus, it is expected that the low virulence strain of triple mutant induces a lower immune response (VN antibody and gB-specific antibody), a delay in the production of VN antibody, and had lower protection rates than the double mutant.

Although no cases of neurological disease caused by BoHV-5 have been reported in China, they have been reported in Europe, the USA, and Australia [41]. However, the increasing importance of China’s cattle industry in the global beef market will probably lead to an increasing number of international trades of cattle and products, which may lead to the entry of BoHV-5 accidentally through animal imports. Therefore, there is a dire need for the development of safer and more effective vaccines against BoHV-5 infection. Previous work in calves suggested that subcutaneous inoculation of either BoHV-1 or BoHV-5 vaccines conferred partial protection to the BoHV-5 challenge [40,42,43]. Given the fact, we checked whether vaccination with BoHV-1 gG-/tk-/gE- and BoHV-1 gG-/tk- would induce cross-protection to BoHV-5 infections. In both groups of mutant-inoculated calves, only partial protection was attained. Nasal BoHV-5 shedding post-challenge was not significantly reduced by vaccination except at 6 dpc. Vaccination also did not fully prevent the development of neurological signs in calves after challenge. However, unvaccinated calves developed more pronounced neurological signs. As a result, both mutants provided insufficient protection to the BoHV-5 challenge, similar to that reported by others [43].

Based on the previous reports regarding the IBR marker vaccine [15] and as elucidated above, the triple mutant BoHV-1 gG-/tk-/gE- may serve as a candidate marker vaccine for controlling BoHV-1 infection in the cattle industry.

## 5. Conclusions

BoHV-1 gG-/tk-/gE- could be a safer candidate marker vaccine against IBR and booster vaccination might be an effective approach to increase the immune response against the challenge, but this requires further investigation. Moreover, the comparative proteomic analysis revealed dysregulation of some unique proteins in this triple mutant BoHV-1 gG-/tk-/gE- and double mutant BoHV-1 gG-/tk- correlated to the wt BoHV-1 that can be used as diagnostic biomarkers.

## Figures and Tables

**Figure 1 vetsci-08-00253-f001:**
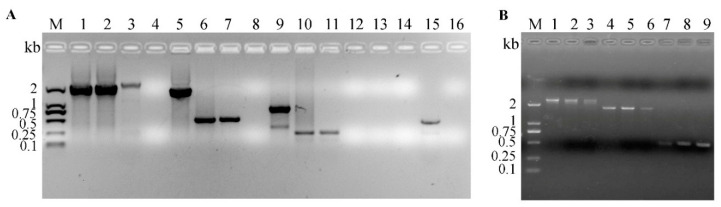
Identification of BoHV-1 gG-/tk-/gE- triple mutant by PCR. (**A**) A fragment of 2183 bp of the intact gE gene fragment in wt BoHV-1 (Lane 1) and BoHV-1 gG-/tk- (Lane 2), and a fragment of 2395 bp including the whole egfp gene and parts of the flanking upstream and downstream sequences (Lane 3) are shown. In addition, the gG gene fragment in wt BoHV-1 (Lane 5), BoHV-1 gG-/tk- (Lane 6), and BoHV-1 gG-/tk-/gE- (Lane 7), and the tk gene fragment in wt BoHV-1 (Lane 9), BoHV-1 gG-/tk- (Lane 10), and BoHV-1 gG-/tk-/gE- (Lane 11) are shown. A fragment of 481 bp including parts of upstream homologous arm gene and egfp gene was detected in BoHV-1 gG-/tk-/gE- (Lane 15), but not in wt BoHV-1 (Lane 13), BoHV-1 gG-/tk- (Lane 14). Lanes 4, 8, 12, 16 represent negative control. (**B**) A fragment of 1685 bp including the whole upstream homologous arm gene, parts of the flanking upstream sequences of the upstream homologous arm gene, and egfp gene were amplified from BoHV-1 gG-/tk-/gE- mutant (Lane 4–6). M represents DNA ladder DL2000.

**Figure 2 vetsci-08-00253-f002:**
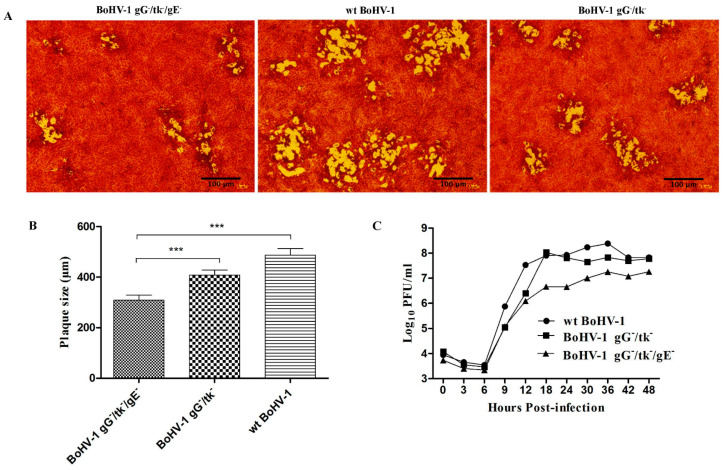
Growth characteristics of BoHV-1 gG-/tk-/gE- compared with wt virus and BoHV-1 gG-/tk- in vitro. (**A**) Plaque morphology of BoHV-1 gG-/tk-/gE-, BoHV-1 gG-/tk-, and wt BoHV-1 in MDBK cell monolayers. (**B**) The diameter of 30 plaques for each virus. (**C**) One-step growth curve of BoHV-1 gG-/tk-/gE- and wt BoHV-1 viruses in MDBK cells. The mark (***) with *p* < 0.001 refers to highly significant.

**Figure 3 vetsci-08-00253-f003:**
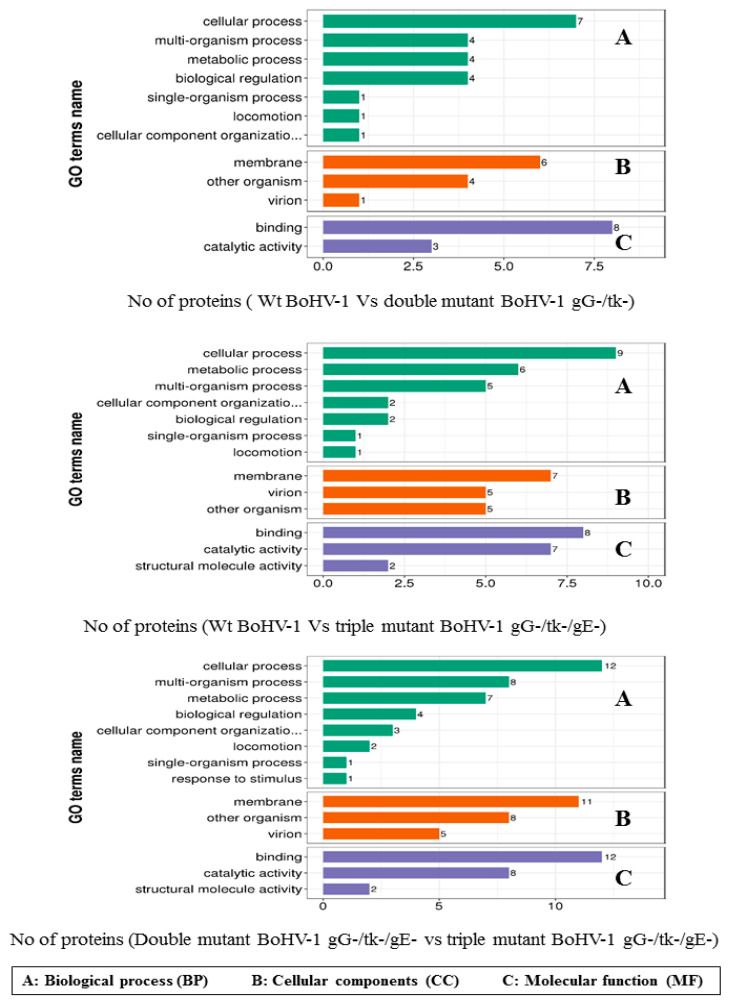
Biological process, cellular components, and molecular function categories of proteomic data by bioinformatic analysis of different comparable groups based on information provided by the online resource Gene Ontology, UniProt-GOA database and the InterProScan software. (**A**) Biological process. (**B**) Cellular components. (**C**) Molecular function.

**Figure 4 vetsci-08-00253-f004:**
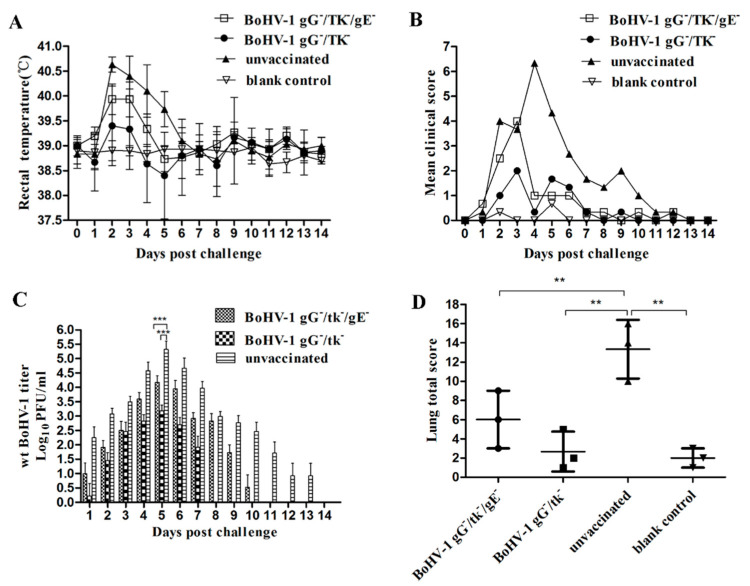
Protective efficacy of BoHV-1 gG-/tk-/gE- against wt BoHV-1 (**A**–**D**) challenge in calves. (**A**) Temperature change. (**B**) Clinical scores. (**C**) Nasal virus shedding. (**D**) Lesion scores of lungs. The mark (**) with *p* < 0.01 and (***) with *p* < 0.001 were considered to indicate a significant or high significant statistical difference respectively.

**Figure 5 vetsci-08-00253-f005:**
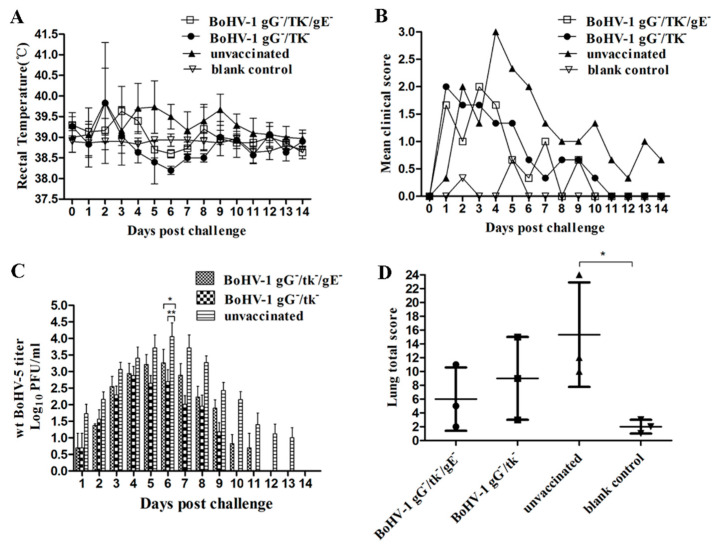
Protective efficacy of BoHV-1 gG-/tk-/gE- against wt BoHV-5 (**A**–**D**) challenge in calves. (**A**) Temperature change. (**B**) Clinical scores. (**C**) Nasal virus shedding. (**D**) Lesion scores of lungs. The mark (*) with *p* < 0.05 and (**) with *p* < 0.01 were considered to indicate a significant difference.

**Table 1 vetsci-08-00253-t001:** Experimental design.

Animal	Virus	No	Vaccinated Route and Dose	Dex Injection ^a^	Challenge Virus	Challenge Date and Dose	Euthanasia and Necropsy
Calf	BoHV-1 gG-/tk-/gE-	6	IN, 4 × 10^7^ PFU	21–25 dpi	wt BoHV-1 (3) ^b^ wt BoHV-5 (3) ^b^	35 dpi, 4 × 10^7^ PFU	28 dpc
	BoHV-1 gG-/tk-	6	IN, 4 × 10^7^ PFU	21–25 dpi	wt BoHV-1 (3) ^b^ wt BoHV-5 (3) ^b^	35 dpi, 4 × 10^7^ PFU	28 dpc
	wt BoHV-1	3	IN, 4 × 10^7^ PFU	21–25 dpi		35 dpi, 4 × 10^7^ PFU	28 dpc
	Unvaccinated control	6	DMEM	21–25 dpi	wt BoHV-1 (3) ^b^ wt BoHV-5 (3) ^b^	35 dpi, 4×10^7^ PFU	28 dpc
	Negative control	3					28 dpc

^a^: Dexamethasone (dex) at 0.1 mg/kg body weight (bwt) was injected intramuscularly into the calves for 5 consecutive days to reactivate the putatively latent virus. ^b^: Six vaccinated calves were allotted randomly to two sub-groups, and three calves in each sub-group were challenged with wt BoHV-1 and wt BoHV-5, respectively. The dpi refers to days post-infection while the dpc refers to days post-challenge.

## Data Availability

Data sharing not applicable.

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
