# Peer review of "Characterization of BoHV-1 gG-/tk-/gE- Mutant in Differential Protein Expression, Virulence, and Immunity"

_vetsci, 2021, doi:10.3390/vetsci8110253_

Round 1
Reviewer 1 Report
The authors describe a triple mutant BoHV-1 in which gG/tk/gE have been deleted. Proteomic analysis identified common up and down regulated virulence proteins to a formerly defined double mutant; in vitro assay showed smaller plaque and slower growth kinetics and in vivo assay showed potential for use as a vaccine candidate, although surprisingly, not as sufficient as the double mutant previously described.
Throughout the methods--10^7 PFU?
Line 101--why is 2% serum being used here when previously cells were grown in 10% FCS?
Figure 2C please include error bars and appropriate stats
Line 263 and 270: "down" regulation
In section 3.3.2 GO Secondary annotation classification: when authors are describing cellular classification, what is meant by 'other organisms'? Clarification or definition of this terminology needs to be included. The authors mention the multiple overlapping of up/down regulated genes but never discuss the importance of these in detail--how would losing/gaining these function help to maintain immunogenicity?
Please label S4/S5 figure 4 etc "ND or none detected" where appropriate
Please include a higher quality image of all figures and use "mock" instead of "blank"
What is "unvaccinated control" in S5? Does this mean "unchallenged" as the animals experience no clinical score or fever?
Throughout the manuscript, more experimental detail is preferred--How many animals were used (simply n=X) and which statistical tests were used for each analysis ("p<0.001, students t-test", for example). Closer attention to detail in terms of grammar should be payed: in many sections there are multiple unnecessary spaces or typos and throughout the figure/tables the font switches to serif/sans serif. Further, since the GO figures are in color, perhaps it would help for many of the column graphs to be colored as well; similarly, making the symbols on XY-graphs larger would be appreciated.
Author Response
Thank you so much dear reviewer for your valuable comments. All comments have been fixed in the attached file

Reviewer 2 Report
The authors present an interesting study which describes the construction, in vitro chracterisation and in vivo characterisation of a triple gene deletion mutant of bovine alphaherpesvirus 1 (BoHV-1). The in vivo evaluation assessed the capacity of the triple mutant to act as a vaccine for homologous challenge against a wild-type strain of BoHV-1 and heterologous challenge with a wild-type strain of bovine alphaherpesvirus 5 (BoHV-1). The manuscript presents a large and comprehensive dataset which support the conclusions drawn by the authors. The manuscript will be of interest to those working on this important virus and the diseases it is associated with. In general resolution of the figures is poor. As this was not evident for the supplemental figures, I suspect this loss of resolution has occurred during the conversion to PDF. Generally, the discussion is quite long and, in many places, paraphrases the results. I would encourage the authors to review this section and try to make it more focused. Keeping it mind it is not essentially to discuss all results. Just those that conflict with or add to the body of scientific literature of the associated field. Two important issues which are identified in the discussion but not really discussed are: 1. Is the triple deletion virus really suitable for use as a vaccine? Some of the evidence provided suggests it may be too attenuated to provide sufficient protection. How would the authors respond to this? 2. In the heterologous (BoHV-5) challenge of animals vaccinated with the triple deletion mutant, the efficacy estimate (68.7%) was higher than the estimate for homologous (BoHV-1) challenge (64.2%). In comparison the double deletion mutant yielded efficacy estimates of 97.1% and 47.2% for homologous and heterologous challenge, respectively. How do these comparative estimates compare to other published studies? What might be driving this effect? I am not convinced of the value of the rabbit experiments described in this study. I presume they were done as model to evaluate if cattle experiments were justifiable. However, in the current version of the manuscript where the cattle experiments (i.e. the natural host) are described it seems the rabbit data adds little to the overall story. Consider removing. Comments and suggestions Line 23 suggest revision “bovine alphaherpesvirus 1” – in line with the latest recommendations of the ICTV. Line 24 suggest replacing “disasters” with “losses” Line 28 The authors state “On proteomic level, it revealed downregulations of some virulence related proteins including thymidine kinase, glycoproteins G, E, I and K when compared to the wildtype. I would suggest that the authors deleted this sentence and consider removing most, if not all, of the proteomics data. It is hardly surprising that the proteomics suggest down-regulation of the three genes which have been deleted from the virus genome. Consequently, the proteomics data is just supporting the conclusions that these genes have been deleted. While the authors have extended these datasets to explore the impacts on the levels of other BoHV-1 proteins, to me this an offshoot of the study, not the main focus which is the capacity of the triple deletion mutant to act as a vaccine strain. As a consequence of including the proteomics data (eg Fig. 3) in the manuscript important and (in my opinion) far more interesting data are relegated to supplemental files. As an example, the estimates of protective efficacy of the triple deletion mutant are relegated to supplemental Table S6 when arguable these estimates were the whole point of the study. Line 46 suggest revision “Herpesviridae” Suggest deletion of “(α-HVs)” – it does not appear to be used in the text and it would be better to use the general term for the subfamily, alphaherpesvirus(es), if required. Formal taxonomic names should also be in italicised text: Varicellovirus, Alphaherpesvirinae Line 49-50 The authors state “Several antigenically similar subtypes of BoHV-1 have been revealed depending on the genomic analysis and viral peptide patterns like BoHV-1.1, BoHV-1.2 and BoHV-1.3” I believe it is now well accepted that there are three genotypes of BoHV-1 namely, BoHV-1.1, BoHV-1.2a and BoHV-1.2b. It is several years since BoHV-1.3 was formally recognised and designated as the separated species bovine alphaherpesvirus 5 (BoHV-5). I do not see any point in perpetuating this outdated nomenclature in the current study. Please revise. Line 55 suggest revision “(IPB), excluding abortion” Line 98 suggest replacing “omitted” with “deleted” Line 100 suggest revision presumable 107 Line 157 Would “avoid mutual interference” be better described as “to prevent intergroup transmission”? Line 165 Please review the viral titres in columns 4, 5,6 & 7, appears the superscripts are not formatted correctly for the viral titres (as per line 100) and letters for footnotes. Line 220-236 Figure 1 legend – it is not necessary to repeat the methodology used in such detail in figure legends. Same comment for other figure legends. Line 237 suggest replacing “determination” with “estimation” Line 238 suggest revision “by fewer and smaller” Line 240 I would suggest that plaque size estimates be rounded to either whole numbers or one decimal place. Line 244 suggest revision “mutant at the remaining time points mutant” Line 247 A magnification or scale bar should be provided for this figure. Line 252 Fig. 2C – significant differences in the viral titres are not illustrate on the figure – as described in the text on lines 242 to 245. Lines 365-378 – I am not convinced that the rabbit data adds anything to this study. Consider removing. Line 374 Were the temperature differences significant in Figure S5A? Line 378 The significant differences are not illustrated on Fig S5B. Line 417 Fig 4A – replace “black” with “blank”. Also with this panel and Fig 4E the number of groups and size make virtually impossible to distinguish between the groups. Line 459 I would suggest splitting the BoHV-1 and BoHV-5 challenge/protection data into separate figures. Line 461 – the lung scoring system used should be described in the materials and methods not the figure legend. Line 523 – did the authors examine the nasal swabs for the presence of virus specific sIgA? While I understand the rationale of examining total sIgA when using a modified live viral vaccine delivered intranasal, I am not a great fan of measuring total sIgA (or IgG) as I am not convinced it provides meaningful data with respect to vaccine efficacy. Line 528 What does “IL-2 was not detectable from any calf.” refer to? Unvaccinated calves? Line 543 I am not sure what the statement “LFQP serves as a gold potential approach” means. Please review and amend as required. However, as stated previously I am not convinced of the value this data to the current study. Line 548 suggest revision “with fewer plaques of reduced size observed during the in vitro experiments.” Line 574 I would have thought that is quite clear that the “gene deletions” have affected virus replication as this what the reported data suggests. Line 596 suggest revision “for controlling BoHV-1 infections in the cattle industry.” Line 600 The authors could consider very briefly outlining what these “further investigations” might involve. Supplemental files Supplemental Fig S2 centrifugation speeds should be quote as gravitational force. Supplemental Fig S5 – consider adding symbols to denote significant differences in Panels A & B. Supplemental Table S7 – Row 1 should be incorporated into the table title. Also I gather the values shown are for the number of animals where IL-4 was detected? Given the number of animals in each group is fixed, this value could be added to column A, then additional columns added for each day with the number of positive animals. The units should also be added for IL-4.Author Response
Thank you so much dear reviewer for your valuable comments. All comments have been fixed in the attached file

Round 2
Reviewer 1 Report
Accept current revised manuscript
Author Response
Thank you dear reviewer for your appreciated and scientific comments. All the comments had been carefully fixed. Thank you so much
